# Lead Decreases Bone Morphogenetic Protein-7 (BMP-7) Expression and Increases Renal Cell Carcinoma Growth in a Sex-Divergent Manner

**DOI:** 10.3390/ijms25116139

**Published:** 2024-06-02

**Authors:** Elizabeth A. Grunz, Haley Anderson, Rebecka M. Ernst, Spencer Price, D’Artanyan Good, Victoria Vieira-Potter, Alan R. Parrish

**Affiliations:** 1Department of Medical Pharmacology and Physiology, School of Medicine, University of Missouri, Columbia, MO 65201, USA; 2Department of Nutrition and Exercise Physiology, College of Agriculture, Food and Natural Resources, University of Missouri, Columbia, MO 65201, USA

**Keywords:** androgen receptor, BMP-7, estrogen receptor α, lead, renal cell carcinoma

## Abstract

Both tissue and blood lead levels are elevated in renal cell carcinoma (RCC) patients. These studies assessed the impact of the subchronic lead challenge on the progression of RCC in vitro and in vivo. Lead challenge of Renca cells with 0.5 μM lead acetate for 10 consecutive passages decreased E-cadherin expression and cell aggregation. Proliferation, colony formation, and wound healing were increased. When lead-challenged cells were injected into mice, tumor size at day 21 was increased; interestingly, this increase was seen in male but not female mice. When mice were challenged with 32 ppm lead in drinking water for 20 weeks prior to tumor cell injection, there was an increase in tumor size in male, but not female, mice at day 21. To investigate the mechanism underlying the sex differences, the expression of sex hormone receptors in Renca cells was examined. Control Renca cells expressed estrogen receptor (ER) alpha but not ER beta or androgen receptor (AR), as assessed by qPCR, and the expression of ERα was increased in tumors in both sexes. In tumor samples harvested from lead-challenged cells, both ERα and AR were detected by qPCR, yet there was a significant decrease in AR seen in lead-challenged tumor cells from male mice only. This was paralleled by a plate-based array demonstrating the same sex difference in BMP-7 gene expression, which was also significantly decreased in tumors harvested from male but not female mice; this finding was validated by immunohistochemistry. A similar expression pattern was seen in tumors harvested from the mice challenged with lead in the drinking water. These data suggest that lead promotes RCC progression in a sex-dependent via a mechanism that may involve sex-divergent changes in BMP-7 expression.

## 1. Introduction

Kidney cancer is the eighth leading cause of cancer and cancer deaths [1]. In 2017, it was predicted that there would be 63,900 new cases and 14,400; over the past decade, new cases have increased by 0.7% annually [1]. Metastasis is a major clinical issue; 16% of new cases have metastasized at diagnosis, and the 5-year survival is 11.7% compared to 92.6% and 66.7% for those with localized and regional tumors, respectively [1]. Renal cell carcinoma (RCC) accounts for 80% of kidney cancers [2]. There are over 10 histological and molecular subtypes of RCC; the most common are clear cell RCC (ccRCC; 75% of cases), papillary RCC (pRCC; 10–15% of cases) and chromophobe RCC (chRCC; 5% of cases). The remaining subtypes are very rare, i.e., less than 1% incidence. Major risk factors for RCC include body weight, elevated blood pressure, and smoking [3]. While RCC is more common in males over 60 years of age [4], the mechanism(s) for this sex difference remains unknown.

Lead is a pervasive environmental contaminant; at low, chronic exposures, decreased renal function is an important concern [5,6], particularly in patients with underlying kidney dysfunction secondary to diabetes or hypertension [7,8]. Human exposure to lead may occur by occupational exposure, ingestion, and smoking, including secondhand smoke [9,10,11]; smoking is a risk factor for RCC [12]. Elevated blood levels of lead are associated with RCC [13,14], and lead accumulation within tumor tissue correlates with advanced RCC stage [15]. In a study of 33 male patients (non-smokers) with RCC, blood levels of lead were 4.90-fold higher than in control patients, respectively [16]. These data suggest that lead may have a role in RCC initiation and/or progression, yet the mechanisms are not known.

Interestingly, there is a sex bias associated with kidney cancer—men are twice as likely to develop cancer than women and have larger and more aggressive tumors [17]. The incidence ratio of 2:1 male/female is consistent across the world [18]. Despite this well-established difference, the pathways underlying the sex bias are not clear. A sex bias has also been observed in experimental models; male mice developed more tumors than female mice in the conditional triple-mutant GEMM model, inactivating *Vhl*, *Trp53*, and *Rb1* [19]. The role of androgen receptor (AR) in renal cell carcinoma is unclear– data suggests that the presence of AR is protective in RCC [20,21], while other studies indicate that increased AR expression is associated with increased proliferation and tumor stage [22,23]. A meta-analysis of 1447 patients indicated that AR expression is 28.2% and correlated with low tumor grade and stage [24]. Estrogen has differential effects on cancer progression depending on cancer type and receptor expression. There are two main estrogen receptors that are studied in cancer biology: ERα and ERβ. In general, ERα is considered pro-oncogenic, whereas ERβ may be a tumor suppressor [25]. Similar to the AR, there are conflicting findings regarding the role of ER expression in RCC. There are data suggesting that ERβ expression inhibits RCC [26,27], while other studies indicated that higher ERβ expression is associated with a poor prognosis [28,29]. Interestingly, the independent role of ERα in RCC prognosis has not yet been determined [29].

Bone morphogenetic proteins (BMPs) are a group of cytokines belonging to the transforming growth factor-β (TGF-β) superfamily [30]. BMPs were initially identified as regulators of bone morphogenesis but also have critical roles in embryonic development, tissue differentiation, metabolism, and cell proliferation [31,32,33,34]. BMPs are also involved in cancer development and progression [35,36]. BMP-7 is the predominant family member involved in kidney development [37] and has been shown to inhibit kidney disease progression in several experimental models [38,39]. In tumor progression, BMP-7 inhibits tumor epithelial-to-mesenchymal transition (EMT) and TGF-β-mediated cell migration and invasion [40,41]. Exogenous BMP-7 decreases proliferation and metastasis of prostate cancer cells [42]. BMP-7 also inhibits tumor growth in an orthotopic xenograft model of breast cancer [43]. In RCC, BMP-7 expression has been shown to be decreased, and loss of expression is associated with the progression of RCC and poor survival [44,45]. BMP-7 has been shown to inhibit cell proliferation of G-402 kidney tumor cells and tumor size in vivo [46]. Sex specificity of the effects of BMP-7 on RCC are not known, yet BMP-7 is regulated by estrogen, which may inhibit its transcription [47].

The current studies were designed to assess the potential role of lead in the progression of RCC to a more aggressive phenotype. In vitro challenge of Renca cells with lead elicited molecular and phenotypic changes that correlate with RCC progression. When lead-challenged cells were injected into mice, tumor size was increased. Interestingly, this was significant in male, but not female, mice; similar sex-divergent results were also seen in mice challenged with lead in drinking water. The sex-divergent effect on tumor size may be related to decreased AR in male mice and/or the observed sex-divergent impact of lead on BMP-7 expression. Taken together, the results support a role for lead in RCC progression and suggest that, mechanistically, lead effects on AR and BMP-7 expression. Furthermore, it appears that these effects are sex-dependent and may ultimately offer some insight into known sex differences in RCC.

## 2. Results

Renca cells were challenged for 10 passages with lead; cells were allowed to seed overnight in complete media (5% FBS) and then challenged for 72 h with 0.5 μM lead acetate (1% FBS) before passage and re-exposure to lead. After 10 passages, a significant reduction in E-cadherin expression was seen, and consistent with the loss of E-cadherin expression, a significant reduction in cell–cell aggregation was observed (Figure 1). Additional phenotypic changes were seen following the subchronic lead challenge. Lead induced an increase in cell proliferation (Figure 2). In the colony formation assay, lead induced over a 10-fold increase in large colonies (>1000 μm) at day 14. In the scratch assay, lead challenge induced a significant increase in wound healing, which may be due to both proliferation and migration. These in vitro results support the conclusion that leads induce RCC progression.

To determine if lead exposure enhances RCC growth in vivo, two experimental strategies were used: (1) exposure of Renca cells to lead prior to inoculation into mice and tumor growth (direct effect on tumor cells; no lead exposure of mice) and (2) sub-chronic lead exposure (drinking water) of mice (systemic effects of lead; no lead exposure of injected cells). Tumor growth was assessed in mice injected with control and lead-challenged Renca cells. On day 14, there were tumors in 5/12 mice injected with control cells, while there were tumors in 8/12 mice injected with lead-challenged cells; on day 21, there were tumors in 9/12 mice injected with control cells and 11/12 mice injected with lead-challenged cells. A significant increase in tumor size (only assessed in tumor-bearing mice) was seen at day 21; interestingly, the increased tumor size was only seen in male, but not female mice (Figure 3A).

Similar results were seen when mice were exposed systemically. Mice were challenged with lead acetate in the drinking water for 20 weeks (32 ppm), and control Renca cells (not lead-challenged Renca cells) were injected subcutaneously in the heterotopic tumor model. This dosing regimen did not induce renal dysfunction as the glomerular filtration rate was not significantly decreased in either male or female mice as assessed by the FITC-sinistrin method; however, body weight was increased in both sexes. A significant increase in tumor size was not seen at day 21 post-injection in mice. However, again, when the data were analyzed by sex, increased tumor size was seen in male but not female mice (Figure 3B). These data provide in vivo evidence that lead, through both direct effects on tumor cells, as well as systemic effects, induces RCC growth via a sex-divergent mechanism.

Renca cells are derived from male mice. However, there was no difference in tumor growth of control cells in males as compared to females in either model. Interestingly, qPCR analysis of control Renca cells demonstrated ERα (Figure 4A) but not AR or ERβ expression. The expression of ERα was increased in tumors from both male and female mice, but there was no sex difference in expression (Figure 4B). In tumors harvested from mice injected with lead-challenged cells, AR and ERα expression was detected by qPCR, and there was a sex-divergent impact on AR expression (Figure 4C). A similar pattern of decreased expression of ERα in tumors harvested from male mice was seen, but the difference was not significant.

To identify molecular pathways that could account for the sex bias of lead-induced tumor growth, RNA was harvested from male and female mice from control and lead-treated tumor cells (direct effect). Gene expression was assessed using the RT^2^ Profiler™ Epithelial to Mesenchymal Transition array. The impact of lead on the expression of BMP-7 was sex-divergent—in males, there was decreased expression in the tumors from the in vitro lead-challenged Renca cells, while expression was increased in tumors harvested from female mice (Figure 5). Immunohistochemical analysis of BMP-7 in the tumors validated these results (Figure 5). Moreover, in the lead-drinking water experiments, BMP-7 was significantly decreased in male mice, but not changed in female mice, further supporting the sex difference in lead-induced BMP-7 expression (Figure 5).

## 3. Discussion

In summary, the data indicate that lead induces the progression of RCC, including increased cell proliferation and tumor growth. While the current study did not address the role of lead in RCC initiation, data support the hypothesis that lead exposure is a risk factor for RCC. The data also indicate that lead promotes the progression of RCC in a sex-divergent manner; this effect was seen in tumors generated from lead-challenged cells and in tumors derived from control cells in mice challenged with lead in the drinking water.

Having established a role for lead in RCC progression, our studies addressed the sex-divergent effect. Sarcomatoid elements—characterized, in part, by the loss of the epithelial phenotype—are prevalent in high-grade RCC [48]; sarcomatoid is associated with all RCC subtypes [49], and EMT is thought to be responsible for the presence of sarcomatoid elements [50]. Given the lead decreased E-cadherin expression in Renca cells, and that loss of E-cadherin is linked to EMT [51], we first examined E-cadherin expression. While decreased E-cadherin expression is seen in tumors generated from the lead-challenged Renca cells, consistent with the loss of expression in the in vitro experiments, there was no sex difference in E-cadherin expression. While lead decreased E-cadherin expression, there was no expression of N-cadherin, vimentin, or a-smooth muscle actin by Western blot analysis, suggesting that lead did not induce a complete EMT phenotypic switch. Additionally, there was no loss of E-cadherin expression in tumors from male or female mice challenged with lead in the drinking water. These data suggest that loss of E-cadherin is not requisite for lead-induced tumor growth or responsible for the sex-divergent effect.

Lead toxicity has a sex bias in immunotoxicity [52] and neurotoxicity [53,54]; however, the mechanism(s) are still being elucidated. Notably, one of the most well-documented effects of lead is a decrease in testosterone [55,56,57,58] and AR levels [55,59]. Prenatal exposure to lead enhances some, but not all, estrogen-induced effects in the female reproductive organs [60,61], suggesting a cell-specific action of lead on estrogen signaling as well. The data suggest that the impact of lead in RCC progression may involve the down-regulation of AR, consistent with several studies suggesting that the presence of AR is protective in RCC [20,21]. There are no lead or sex effects on ERα expression, and ERβ was not detected in Renca cells or tumor tissue. The data does demonstrate that expression of ERα is low in cells and increased in tumor tissue. An interesting finding was that AR expression was not detected in tumors generated from low passage Renca cells but was only detected in tumors derived from the in vitro lead challenge experiments, suggesting cell passage increased AR expression. The low expression of ERα and AR in Renca cells demonstrates that in vitro cell experiments may not be useful in investigating the role of sex hormones and receptors in RCC progression.

There is evidence that BMP-7 is regulated by sex hormones. BMP-7 expression is greater in the male hippocampus than in females [62]. More directly, BMP-7 mRNA is decreased by orchidectomy and increased by testosterone and dihydrotestosterone in mouse prostate [63]. In addition, there is a positive correlation of BMP-7 with estrogen receptors and progesterone receptors in breast cancer [64]. As such, 17β-estradiol increases BMP-7 mRNA expression in bone marrow mesenchymal stem cells [65]. Interestingly, downregulation of BMP-7 was seen in male, but not female, mice in a bladder outlet obstruction model [66], which correlates with sex-dependent fibrosis and dysfunction [67]. The data suggest that the downregulation of BMP-7 may be mechanistically linked to the lead-induced decrease in AR expression.

In summary, the data provide evidence for the role of lead in RCC progression. Importantly, this effect appears to be sex-divergent. We hypothesize that an alteration in the balance of signaling via AR and ERα may enhance RCC progression, perhaps in a sex-specific manner. These data provide a strong rationale for future studies examining the mechanism of lead-induced decreases in the expression of AR and BMP-7, as well as sex differences in such mechanism(s) and the role of these changes in increased RCC tumor growth.

## 4. Materials and Methods

### 4.1. Cell Lines

Renca (mouse; ATTC^®^ CRL-2947) cells were purchased from ATTC and cultured in RPMI 1640 supplemented with 0.1 mM non-essential amino acids, 1 mM sodium pyruvate, 2 mM L-glutamine, 5% FBessence (Avantor Seradigm) and pen-strep (50 U/mL and 50 mg/mL, respectively). For the subchronic challenge, cells were plated at 1 × 10^4^ cells/cm^2^ and allowed to attach overnight in RPMI+5% FBessence. Cells were challenged with 0.5 μm lead acetate in RPMI+1% FBessence for 72 h; this protocol was repeated for 10 passages.

### 4.2. Western Blot

Cells were washed twice with ice-cold 1× DPBS before the addition of lysis buffer (10 mM Tris-HCl, 1% SDS) containing Halt^TM^ Protease/Phosphatase inhibitors (Thermo Scientific, Waltham, MA, USA). Cells were scraped and incubated on a rocker for 15 min at 4 °C. The lysate was pipetted 15 times and spun at 800 rpm for 5 min at 4 °C. Tumor cell lysates were generated by homogenizing tissue in lysis buffer. Protein concentration was determined by the Pierce™ BCA protein assay kit (Thermo Scientific). Total cellular protein was separated on a 4–20% mini-PROTEAN^®^ gel (Bio-Rad, Hercules CA, USA) and transferred onto an Amersham™ Hybond™ PVDF membrane (GE Healthcare Life Sciences, Chicago, IL, USA).

The following antibodies were used: anti-E-cadherin (BD Transduction Laboratories™ Cat #610182), AR (rabbit polyclonal; Invitrogen, Waltham, MA, USA), ERα (Novus, Tokyo, Japan) and anti-β-actin (Sigma, Kawasaki, Kanagawa); all primary antibody dilutions were 1:1000. Goat-anti-mouse or anti-rabbit HRP conjugate (Jackson ImmunoResearch Laboratories, Cat #115035003 and 305035003, West Grove, PA, USA) was used at 1:5000 dilutions. Blots were developed using SuperSignal West Femto Chemiluminescent Substrate (Therm Scientific, Waltham, MA, USA) and imaged using the ChemiDoc^TM^ imaging system (Bio-Rad, Hercules, CA, USA).

### 4.3. Cell Aggregation

Cells were incubated in 5 mL Moscona’s Low Bicarbonate Buffer (MLB) containing 2.5 mM EDTA for 10 min at RT and collected by scraping into a 15 mL tube. Plates were washed in 5 mL MLB, added to the tube, and pelleted (1000× *g*, 5 min). Cells were washed a second time with 5 mL MLB, counted, pelleted, and suspended to a final concentration of 5 × 10^5^ cells/mL in MLB/3 mM CaCl_2_/4 mM MgCl_2_. Twenty-four-well plates were pretreated with 500 µl MLB/1% BSA for 20 min and air-dried for 20–30 min. An amount of 100 µl of cells (5 × 10^4^) + 100 µl MLB/1% BSA was added to each well for a final concentration of MLB/1.5 mM CaCl_2_/2 mM MgCl_2_/0.5% BSA. In certain experiments, EDTA was added to a final concentration of 2 mM. Plates were incubated at 37 °C and visualized at 10× magnification at 2 h; clusters of >5 cells were counted.

### 4.4. Proliferation

Cells were seeded at 1 × $10^{4}$ cells/${cm}^{2}$ in 96 well plates, and cell proliferation was assessed over a 5-day time course. In each well, cells were fixed with 100 μL of absolute ethanol for 90 s after growth media was aspirated. Following ethanol aspiration, 75 μL of Janus Green was added in each well for 60 s, after which the dye was aspirated, the wells were rinsed twice with distilled water, and 150 μL of absolute ethanol was added to extract the dye from the cells. Plates were placed on a rocker for 10 min. Cell proliferation at each time point was assessed using Janus Green spectrophotometric assay and absorbance read at 654 nm on the Synergy HT Multi-Detection Microplate Reader for quantification; growth is represented as fold-increase compared to Day 1.

### 4.5. Colony Formation

Cells were seeded in 24 well plates at a density of 200 cells/ml. At each time point (Day 7, 10, and 14), cells were fixed using 2% paraformaldehyde for 5 min, washed twice with distilled water, and stained for 5 min using 0.25% crystal violet, after which the wells were washed twice with distilled water, and each cell line was viewed under the inverted microscope at 4× magnification and number of colonies were counted.

### 4.6. Wound Healing

Cells were seeded in 12-well plates. After 48 h in culture, a horizontal scratch was made using a 10 μL pipetman tip. Media were removed, and the wells were washed 2× with serum-free media. Cells were then incubated with 5% FBEssence for 48 h; cells were then fixed with 95% ethanol and stained with crystal violet. To quantify healing, the center area of the scratch was measured using the closed polygon tool (cellSense; Olympus, Shinjuku City, Japan), and data were presented as %control (T0).

### 4.7. Heterotopic Tumors

All animal use and experimental procedures were approved by the ACUC of the University of Missouri (Protocol 18062). Animals were used in accordance with the IACUC guidelines. Albino BALB/cJ mice (stock number: 000651) were purchased from Jackson Laboratories. Both males and females were aged 8 weeks. Mice were kept in enclosures with littermates unless separation was required for the safety of the animals and fed a standard diet. Mice were injected subcutaneously in the hind flank region with 100 μL of cell solution, containing approximately 10^5^ Renca cells suspended in PBS. Tumors were allowed to grow for 3–4 weeks; tumors were measured with a caliper every seven days. Mice in experimental groups were provided water with 32 ppm of lead (II) acetate (Pb(CH3COO)_2_), which they consumed ad libitum for up to 20 weeks as previously described [68] before injections with control Renca cells; this dosing regimen was associated with blood lead levels of 32.1 ± 11.4 μg/dL [68].

### 4.8. Plate-Based Array/qPCR

For plate-based arrays, total RNA from frozen kidneys was extracted using the NucleoSpin RNA kit with on-column DNase digestion (Macherey-Nagel, Düren, Germany), and RNA yield and purity were determined using a NanoDrop spectrophotometer (ThermoFisher, Waltham, MA, USA). cDNA was synthesized using the RT^2^ First Strand Kit (Qiagen, Hong Kong), in which 0.5 µg of total RNA was used for reverse transcription of each sample. The mouse epithelial-to-mesenchymal transition (EMT) RT^2^ Profiler PCR Array (Qiagen) was run using RT^2^ SYBR green mastermix (Qiagen) on a CFX96 Touch Real-Time PCR machine (Bio-Rad). Cycling conditions proceeded with a 10 min incubation at 95 °C, followed by 40 amplification cycles at 95 °C for 15 s and 60 °C for 1 min. A melt curve was run to verify PCR specificity. C_T_ values were uploaded to the GeneGlobe Data Analysis Center (Qiagen) for analysis.

For qPCR, total RNA from frozen tumor samples or Renca cells was extracted using the NucleoSpin RNA kit with on-column DNase digestion (Macherey-Nagel, Waltham, MA, USA), and RNA yield and purity were determined using a NanoDrop spectrophotometer. cDNA was synthesized using a high-capacity cDNA reverse transcription kit (Applied Biosystems, Waltham, MA, USA), including an RNase inhibitor. Bullseye multi ROX TaqProbe qPCR Master Mix (MidSci, St. Louis, MI, USA), nuclease-free water, and TaqMan gene expression assays (ThermoFisher)) were used in creating assay-specific premixes. The recommended mouse TaqMan assay was purchased in the FAM-MGB dye for each of the following: Casc3 (Mm01296308_m1), AR (Mm00442688_m1), Esr1 (Mm00433149_m1), Esr2 (Mm00599821_m1), and Pgr (Mm00435628_m1). Amplification was performed using 100 ng cDNA per sample. Reactions were carried out in a 96-well optical reaction plate using a CFX96 Touch Real-Time PCR machine (Bio-Rad). Thermal conditions proceeded with an initial 20 s incubation at 95 °C followed by 40 cycles of amplification: 95 °C for 1 s and 60 °C for 20 s. Fold changes in gene expression were calculated using the C_T_ method [69]. The expression of target genes was normalized by comparing the raw C_T_ values of the samples to those of the Casc3 reference gene.

### 4.9. Immunohistochemistry

Slides were deparaffinized and rehydrated according to the following: 12 min xylenes, 5 min 95% EtOH, 5 min 80% EtOH, 5 min 70% EtOH, 5 min 50% EtOH, and 10 min 1× tris-buffered saline (TBS). Slides were stained for BMP-7 using a rabbit polyclonal antibody (1:1000; Thermo Fisher). All slides were viewed on an Olympus IX51 microscope, and images were taken using cellSense Dimension software (v4.1.1) with an automatic exposure time and white balance on all slides at a 10× magnification.

### 4.10. Statistics

Results are expressed as mean + standard deviation. A one-way analysis of variance (ANOVA) was performed, followed by Tukey’s multiple comparison test using the statistical software GraphPad Prism 9 (GraphPad Software, La Jolla, CA, USA). The differences were considered statistically significant at *p* < 0.05.

## Figures and Tables

**Figure 1 ijms-25-06139-f001:**
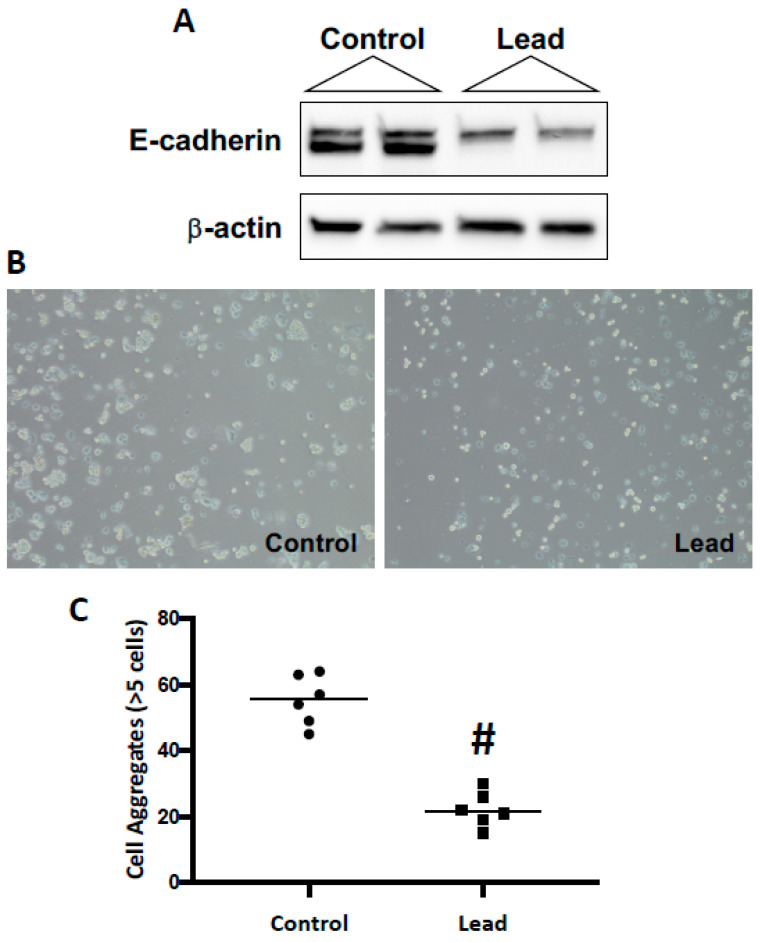
Impact of lead on cell adhesion in Renca cells. Renca cells were challenged with 0.5 μM lead acetate for 10 passages. (**A**): Western blot analysis of E-cadherin in lead-challenged Renca cells, while (**B**) shows cell–cell aggregation in control and lead-challenged Renca cells (10×); the quantitative results are shown in (**C**). # indicates a significant difference from control (*p* < 0.0001).

**Figure 2 ijms-25-06139-f002:**
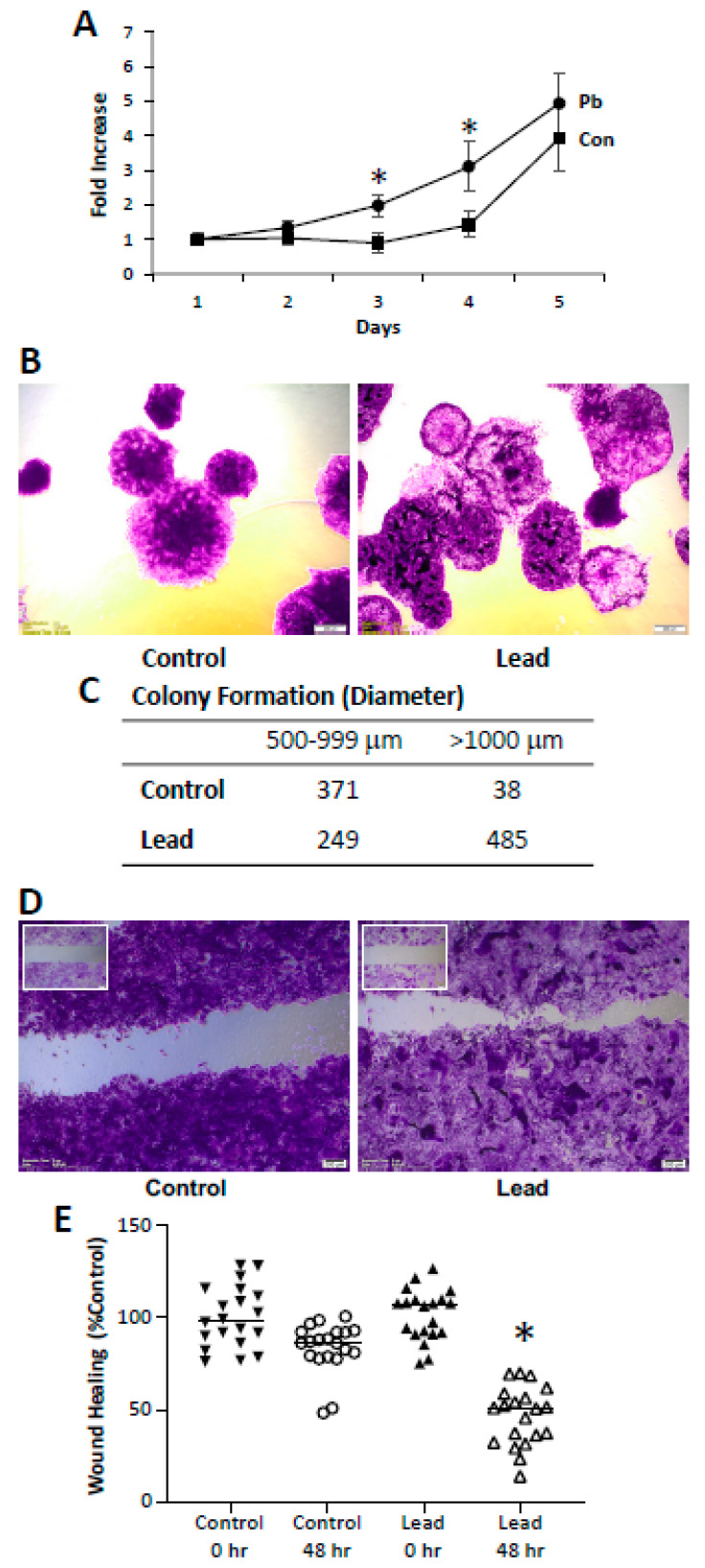
Impact of lead on RCC progression in vitro. Renca cells were challenged with 0.5 μM lead acetate for 10 passages. (**A**) depicts cell proliferation. Colony formation is shown in (**B**, 10×); quantitative results are shown in the table (**C**). Wound healing in the scratch assay at 48 h is shown in (**D**, 10×), and the quantitative results are shown in (**E**). * indicates a significant difference from control (*p* < 0.05).

**Figure 3 ijms-25-06139-f003:**
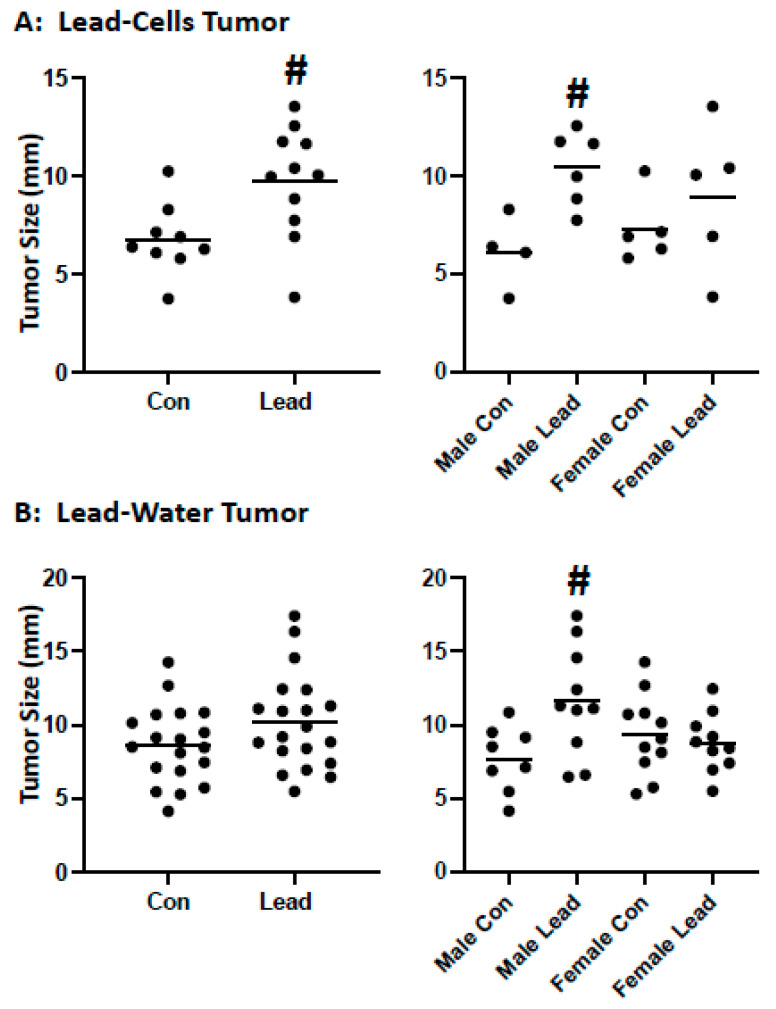
Impact of lead on tumor growth. Lead-Cells Tumor: Renca cells were challenged with 0.5 μM lead acetate for 10 passages prior to subcutaneous injection in the hind flank of male and female Balb/cJ mice. (**A**): Tumor growth at 21 days is shown; lead-challenged cells had increased tumor growth as compared to control cells as indicated # (*p* = 0.0127); the significant difference was confined to male mice (# *p* = 0.0076), but not female mice (*p* = 0.3858). (**B**): Control Renca cells were injected into male and female Balb/cJ mice—control or challenged with 32 ppm lead in drinking water for 20 weeks. There was no significant difference in tumor size when mice were examined as a group (*p* = 0.1125), but a significant increase was seen in male mice (# *p* = 0.015) but not female mice (*p* = 0.5972).

**Figure 4 ijms-25-06139-f004:**
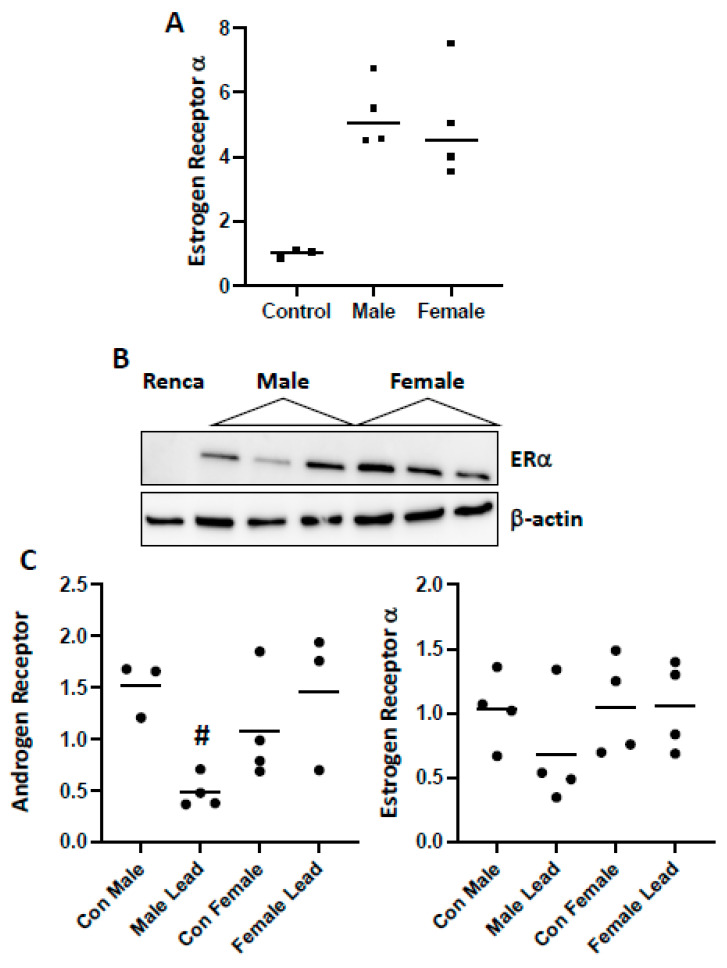
Sex hormone expression in Renca cells and tumors. (**A**): qPCR analysis of ERα expression in control Renca cells and tumors harvested from male and female mice injected with control cells is shown; there is an increase in expression in tumor samples as compared to cells. In (**B**), Western blot analysis demonstrates a similar finding of increased expression in tumor samples as compared to cells. (**C**) depicts qPCR analysis of AR and ERα expression in tumor samples harvested from male and female mice injected with lead-challenged Renca cells. AR expression was decreased in tumors derived from lead-challenged cells harvested from the male (# *p* = 0.0089) but not female mice. There were no sex-divergent effects on ERα expression; although it tended to be lower in tumors derived from lead-challenged Renca cells in males, it was not significant (*p* = 0.2425).

**Figure 5 ijms-25-06139-f005:**
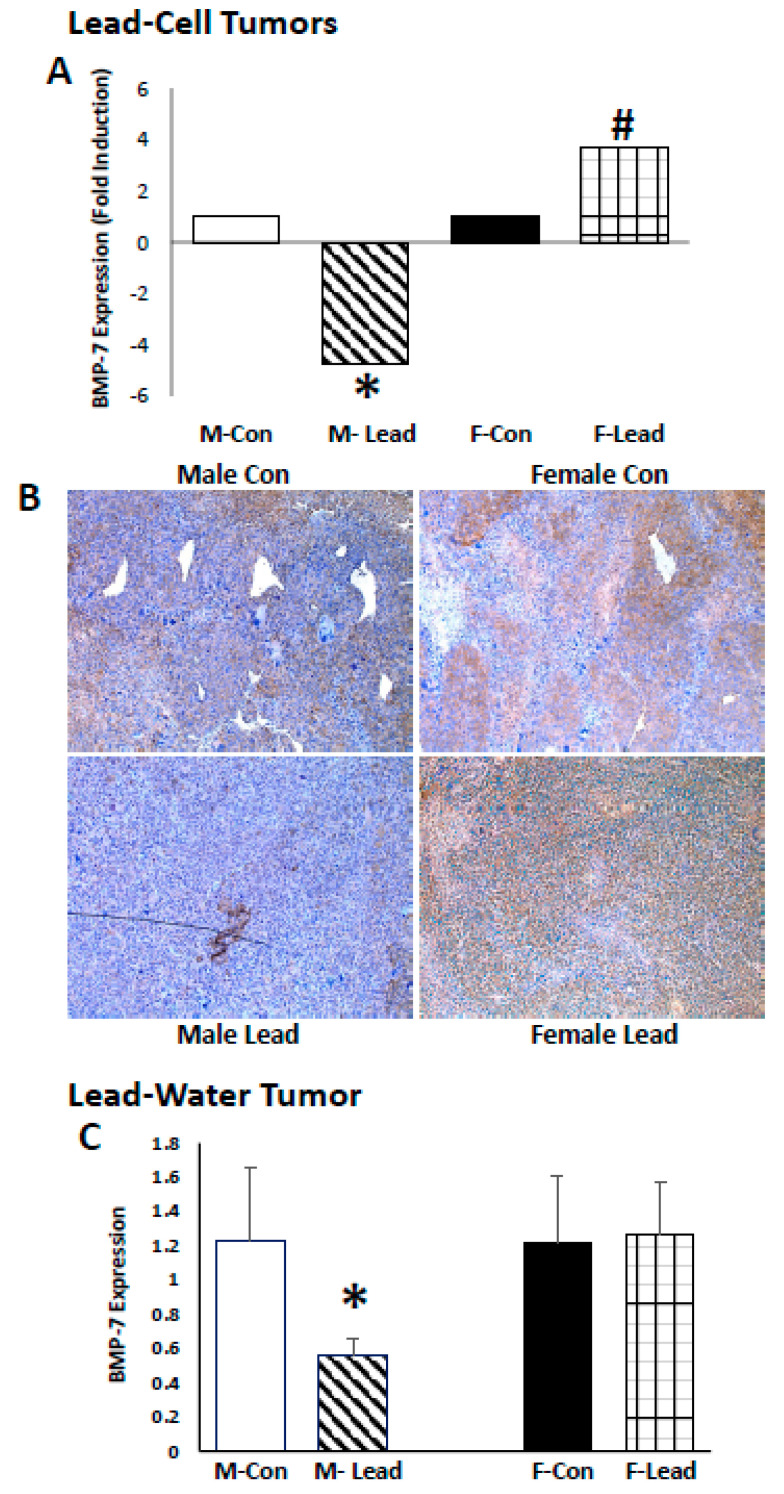
Sex-dependent impact on lead-induced alterations in BMP-7 expression. Renca cells were challenged with 0.5 μM lead acetate for 10 passages prior to subcutaneous injection in the hind flank of male and female Balb/cJ mice. Gene expression (**A**) of three samples per group using the RT^2^ Profiler PCR Gene Array: Mouse EMT. A significant decrease (#) in BMP-7 was observed in tumors harvested from male mice, while a significant increase (#) in expression was seen in tumors harvested from female mice. (**B**): Immunohistochemical analysis demonstrates a decreased expression in tumors harvested from male mice (10×) (**C**): qPCR analysis of BMP-7 expression in tumor samples harvested from the lead drinking water expression is shown; * indicates a significant difference from the respective control.

## Data Availability

The raw data supporting the conclusions of this article will be made available by the authors upon request.

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
