# Peer review of "Lead Decreases Bone Morphogenetic Protein-7 (BMP-7) Expression and Increases Renal Cell Carcinoma Growth in a Sex-Divergent Manner"

_ijms, 2024, doi:10.3390/ijms25116139_

Round 1
Reviewer 1 Report
Comments and Suggestions for Authors
Manuscript # ijms-3012347
Title: Lead Decreases BMP-7 Expression and Increases Renal Cell Carcinoma Growth in a Sex-Divergent Manner
In this study, authors designed to assess the impact of subchronic lead challenge on the progression of RCC in vitro and in vivo. These data suggest that t lead promotes RCC progression in a sex-dependent via a 26 mechanism that may involve sex-divergent changes in BMP-7 expression
1. Introduction, line 47, renal cell carcinoma (RCC)
2. How much concentrations of lead used in this study?, Only one concentration ?
3. The used lead concentration (0.5 mM) is equivalent to human exposure levels in RCC patients?
4. In the line 140, “Mice were chal- 140 lenged with lead acetate in the drinking water for 20 weeks (32 ppm) and control Renca 141 cells were used in the heterotopic tumor model.”
5. Authors must be measured the lead concentrations in the blood or tumor tissues
6. How about the expression levels of BMP-7 in the kidney ?
7. Line 215, androgen 215 receptor (AR)
8. In the development of RCC, what is the functional role of estrogen receptor ?
Comments on the Quality of English Language
No
Author Response
1. Introduction, line 47, renal cell carcinoma (RCC)
We have deleted renal cell carcinoma (line 53/47) and used RCC since it appears previously in the paper (line 36).
2. How much concentrations of lead used this study? Only one concentration?
We have used higher (0.625 and 1 mM) concentrations of lead in the in vitro studies and seen similar phenotypic effects; however, 0.5 mM is the lowest concentration that we have used for the tumor studies.
3. The used lead concentration (0.5 mM) is equivalent to human exposure levels in RCC patients?
The concentration used in our study (0.5 M) at the high end of a normal human exposure to lead 10 mg/dl is 0.48 M (https://www.ucsfhealth.org/medical-tests/lead-levels---blood)
4. In the line 140, “mice were chal- 140 lenged with lead acetate in the drinking water for 20 weeks (32 ppm) and control Renca 141 cells were used in the heterotopic tumor model”.
The revised paper clarifies that only control (not lead-challenged cells) were used in the lead drinking water model.
5. Authors must be measured the lead concentration in the blood or tumor tissues.
We have not yet measure lead levels in blood or tumor tissue; we have added the information that this exposure leads to blood lead levels of 32.1+11.4 g/dl in mice in the Methods section.
6. How about the expression of BMP-7 in the kidney?
While BMP-7 is the predominant family member expressed in the kidney (reference 37; we have not yet evaluated the impact of lead on expression in the kidney in the drinking water experiments.
7. Line 215, androgen 215 receptor (AR).
We have deleted “androgen receptor” in this sentence, since AR has been defined.
8. In the development of RCC, what is the functional role of estrogen receptor?
In the Introduction, conflicting studies on the role of ERb in RCC have been referenced. El-Deek et al. have shown that “The independent role of ER subunits as markers of poor prognosis is proven only for ERβ and ERα36 but not ER”. We have added this information to the third paragraph of the Introduction.
Reviewer 2 Report
Comments and Suggestions for Authors
This communication is an interesting study of the effects of heavy metal (Pb) on renal cell carcinoma, which identified the a sex-divergent manner.
1. More EMT biomarkers could be determined, besides E-cadherin.
2. The pictures of tumor growth need be presented.
3. It will improve the quality of this paper if verify the effects of Pb on progressions of renal cell carcinoma after biomedical or biogenetic regulating the expression of BMP-7.
Author Response
1. More EMT biomarkers could be determined, besides E-cadherin.
The expression of N-cadherin, vimentin and a-smooth muscle actin – markers of EMT – was also examined; expression was not detected in lead-challenged Renca cells or tumors. This data suggests that lead does not induce complete EMT and this information has been added to the second paragraph of the Discussion.
2. The pictures of tumor growth need to be presented.
We did not take pictures of the mice with the tumors in either the lead-challenged cell study or the drinking water study.
3. It will improve the quality of this paper if verify the effects of Pb on progressions of renal cell cancer after biomedical or biogenetic regulating the expression of BMP-7.
We have preliminary data that exogenous BMP-7 (100 ng/ml; 48 hr) inhibits proliferation of control and lead-challenged Renca cells. However, we have not yet repeated this study, and only have a single concentration and timepoint in pilot experiment.